# ‘Health in All Policies’—A Key Driver for Health and Well-Being in a Post-COVID-19 Pandemic World

**DOI:** 10.3390/ijerph18189468

**Published:** 2021-09-08

**Authors:** Liz Green, Kathryn Ashton, Mark A. Bellis, Timo Clemens, Margaret Douglas

**Affiliations:** 1Policy and International Health, WHO Collaborating Centre on ‘Investment in Health and Well-Being’, Public Health Wales, Cardiff CF10 45Z, UK; kathryn.ashton2@wales.nhs.uk (K.A.); mark.bellis@wales.nhs.uk (M.A.B.); 2Department of International Health, Care and Public Health Research Institute—CAPHRI, Maastricht University, Duboisdomein 30, 6229 GT Maastricht, The Netherlands; timo.clemens@maastrichtuniversity.nl; 3Department of Public Health and Life Sciences, Bangor University, Bangor LL57 2DG, UK; 4Usher Institute, University of Edinburgh, Edinburgh EH16 4UX, UK; margaret.douglas@ed.ac.uk

**Keywords:** health in all policies, health impact assessment, health lens analysis, social determinants of health, policy, advocacy

## Abstract

Policy in all sectors affects health, through multiple pathways and determinants. Health in all policies (HiAP) is an approach that seeks to identify and influence the health and equity impacts of policy decisions, to enhance health benefits and avoid harm. This usually involves the use of health impact assessment or health lens analysis. There is growing international experience in these approaches, and some countries have cross-sectoral governance structures that prioritize the assessment of the policies that are most likely to affect health. The fundamental elements of HiAP are inter-sectoral collaboration, policy influence, and holistic consideration of the range of health determinants affected by a policy area or proposal. HiAP requires public health professionals to invest time to build partnerships and engage meaningfully with the sectors affecting the social determinants of health and health equity. With commitment, political will and tools such as the health impact assessment, it provides a powerful approach to integrated policymaking that promotes health, well-being, and equity. The COVID-19 pandemic has raised the profile of public health and highlighted the links between health and other policy areas. This paper describes the rationale for, and principles underpinning, HiAP mechanisms, including HIA, experiences, challenges and opportunities for the future.

## 1. Introduction: ‘Medicine at a Large Scale’

Policies, plans, and decisions formed inside and outside of the health and care sectors all affect health and well-being [1,2,3]. Policies in areas such as spatial planning, transport, the economy, and the environment can have both positive and negative impacts for population health, and can cause, exacerbate, or reduce health inequalities [4,5,6]. These traditionally described ‘non-health’ sectors and settings are core to the socio-economic, cultural and environmental conditions and determinants of health [7]. This means that to improve health and reduce health inequalities, it is essential to engage with the wider impacts of policies and decisions in all sectors. This is the rationale for the concept of ‘health in all policies’ [8,9]. Health in all policies (HiAP) is a defined as ‘an approach to public policies across sectors that systematically takes into account the health and health systems implications of decisions, seeks synergies and avoids harmful health impacts, in order to improve population health and health equity‘ [8].

Internationally, the COVID-19 pandemic has raised the profile of public health and also highlighted the influence of other sectors on not only physical health, but also social, environmental and economic health determinants [10,11,12]. Alongside this, climate change poses an even greater threat to health and the environment, particularly for the most vulnerable populations [13]. These challenges demand integrated responses across many sectors, to mitigate their effects on health and other outcomes.

This narrative review presents the concept of ‘health in all policies’, and the principles, tools, and methods used to implement it. It describes how it can drive policies that offer co-benefits to health, well-being, and other systems, for example, spatial planning and the environment. The purpose of this paper is to highlight the need to reinvigorate the application of HIAP in the context of the COVID-19 pandemic, the recovery from it, and the upcoming challenges to health and equity. The paper draws on two decades of work and the experiences of conducting multiple health impact assessments (HIA) in Wales and Scotland, molding the development of HIA as a mechanism to achieve HiAP in those nations for two decades. It reflects on what HiAP is, and is not, the mechanisms and resources that can support it and the challenges to its implementation. It argues that the public health community should build on its heightened profile, due to the COVID-19 pandemic, and renew and use HiAP approaches to ensure that health becomes a core consideration in future decisions and policymaking.

### Nothing New under the Sun?

HiAP as a concept is not novel. It was first described in the early twenty-first century, when it became the focus of the Finnish EU Presidency in 2006 [14] and was adopted in South Australia in 2007 [15]. However, it draws on the understandings of the determinants of health that date back much further. The public health movement of the nineteenth century recognized the impact of living and working conditions on health, and sought to improve these [8]. The WHO Constitution in 1948 recognized a broad definition of health and identified the need to work with other agencies [16]. The 1978 Declaration of Alma Ata established equity and intersectoral action for health (IAH) as fundamental to achieving health for all [17]. However, IAH has usually meant for partnership projects to address specific health issues, rather than influencing sectoral policies that may affect health through multiple determinants [18]. The 1986 Ottawa Charter identified, as a key action for health promotion, the creation of the healthy public policy, which ‘puts health on the agenda of policymakers in all sectors and at all levels, directing them to be aware of the health consequences of their decisions and to accept their responsibilities for health’ [19]. HIA began to be used as a methodology to achieve healthy public policy from the 1990s [20,21,22,23,24].

HiAP involves building inter-sectoral collaborative relationships, in order to develop healthy public policies. It is built on several core pillars, including capacity building, governance and accountability, joint or shared resourcing, and partnership working [25,26,27,28]. The approach emphasizes the consequences of public policies on health determinants and inequalities, and aims to improve the accountability of policymakers for health and well-being impacts at all levels of policymaking [29]. It aims to improve and protect population health by working collaboratively across sectors to inform and influence evidence-based decisions, so that negative impacts on health and well-being are avoided or mitigated, and positive impacts are enhanced. This can support a whole-of-government approach that creates shared accountability for health and well-being. This shared accountability and commitment has been termed ‘governance for health’ [18].

The concept of HiAP can be interpreted and used in differing ways. It is sometimes implicitly discussed as an outcome, which could be changes to public policies, to maximize the benefits to health and health equity, or even the ultimate outcome of improved population health and reduced health inequalities [30]. It is more often considered to be an approach—a set of processes, tools, and structures that is intended to achieve the outcome of better policy, and better health and well-being. In this paper, the authors focus on the mechanisms that form the HiAP approach, recognizing that their ultimate purpose is healthier public policies. 

## 2. Does HiAP Differ from Other Public Health Approaches?

The fundamental elements of the HiAP approach include inter-sectoral collaboration, policy advocacy aiming to influence policies to improve their impacts, and a holistic consideration of the range of potential health determinants that are affected by each policy area [9]. 

There are several ways that public health professionals engage in inter-sectoral collaboration, not all of which are HiAP. A common example is the development of inter-sectoral plans or projects to address a public health issue, such as working with transport, planning, and other colleagues to increase physical activity through active travel. These projects usually involve the development of initiatives or interventions. HiAP, however, is not an intervention, but a process seeking to influence wider policies [18].

Policy advocacy includes lobbying, campaigning, and engaging with decision and policymakers to address specific public health issues, such as tobacco or alcohol, or to promote specific policy solutions [31]. This is often, though not always, from a position outside the policymaking process [32]. Combining policy advocacy and inter-sectoral collaboration would include ‘whole of government’ approaches to a defined public health problem, such as obesity that is known to have complex pathways and determinants. This can lead to a broad strategy for the public health issue, incorporating policy solutions across multiple sectors. Some authors may define this as HiAP [33]. However, this focus on a single public health issue does not allow a holistic, comprehensive understanding of each sector’s impacts on health, and whilst it is valuable and legitimate public health work, it is not the same as a HiAP approach.

Unlike the approaches discussed above, in a HiAP approach, the starting point and focus is not a single public health issue, but a singular policy area or specific proposed policy [34]. For example, a traditional public health approach may start with a problem such as physical inactivity, and seek to work with a range of partners, such as transport policymakers, whose policies might influence physical activity levels in order to address this. On the other hand, a HiAP approach starts with a policy area, such as transport policy. It then aims to develop a holistic understanding of how the policy area may affect not only physical activity, but a range of relevant health determinants, for example, air quality, injuries, severance, and others, in order to develop a policy that will gain the best overall health and equity outcomes. This is a crucial difference. It means that HiAP work requires a more detailed understanding of the constraints and opportunities of the relevant policy area. Strong working relationships between public health and colleagues in the other policy area are important to facilitate this. It also requires specific mechanisms or tools, such as HIA, to identify and assess the range of potential links with health. Figure 1 illustrates how HiAP differs from other forms of policy advocacy and inter-sectoral collaboration.

## 3. HIAP Mechanisms

At the core of HiAP is collective, integrated working, and multi-disciplinary and multi-sectoral stakeholder collaboration, to identify and address health issues arising from a policy. There are several specific tools or processes that can be used to drive this and implement HiAP in practice. The most commonly applied of these are HIA and health lens analysis (HLA).

HIA is a systematic, flexible and practical process that can be applied to a policy, plan, strategy, or proposal [37,38]. It is usually carried out prospectively before the policy is implemented, in order to influence changes to the proposal that will improve its health impact and reduce inequalities. It routinely involves a five-step process, which identifies the potential positive and negative health and well-being impacts, and the distribution of those impacts across a population [39]. These steps are shown in Table 1, with an illustrative example of a HIA implemented in practice.

HIA is equity-focused and highlights population groups who may be disproportionally affected by the policy being considered [41]. It includes recommendations or suggested future actions to be taken to enhance positive impacts and mitigate negative impacts, particularly for populations with the poorest health [39,41]. HIA has been used in many settings, contexts, and in a wide variety of policy areas [39,40,41,42,43,44]. It can also be applied in ‘real time’ to unexpected important events, such as the COVID-19 pandemic, in order to inform policy responses and actions [39,45,46]. It is a valuable approach for applying HiAP, as it aims to inform decisions, enable collective and synergistic actions, involves key stakeholders, including policymakers, practitioners, and communities, and addresses potential future inequalities [47,48].

HLA differs from HIA in its positioning and timing in the policymaking cycle [49]. It was developed in South Australia to enable a joined-up approach within the government of South Australia [50]. It has been used less in other jurisdictions, but there is some recent experience of its use in North America [51]. HLA starts at the agenda setting and development stage of policymaking [52]. It aims to identify key interactions and synergies between the policy area and health, to develop policies that will benefit both health and other outcomes, lead to co-benefits across systems, or ‘win:wins’ for all [50,53]. As shown in Table 2, similarly to HIA, it is a systematic process and consists of five essential stages or components, which underpin its effectiveness. Table 2 depicts these stages and how they were implemented as part of a HLA in practice.

As the examples above show, there are more similarities than differences between HIA and HLA. Both involve collating evidence from research and stakeholders, to identify and understand a range of health impacts that are relevant to a policy area, and make recommendations to improve health. The main difference is the entry point [54]. A HLA starts early in the policy process and the HLA team is involved in developing policy responses and then gaining approval for them. A HIA is an assessment of a policy proposal or decision that has already been defined (in the example, this was the moratorium on UOG), and the HIA team is not necessarily involved in further policy development after making recommendations. In practice, however, HIA may also be used more flexibly to support a HiAP approach at other stages of the policy cycle. A comparison of HLA and HIA, as used in two Australian states, found that both approaches enabled evidence-based recommendations to develop a policy that improved health and equity [52]. The main difference was in the organizational positioning, rather than the mechanism used. South Australia used HLAs, positioned inside the government, and was better able to influence policy. In New South Wales, HIAs were completed outside of the government, providing more freedom to collaborate with wider partners, and were not restricted by government priorities. However, there are examples of HIAs sitting outside of the government, which have influenced government policy, particularly in Wales [39,46]. 

Both HIA and HLA seek specifically to understand the links between health and other sectors, and to influence policies accordingly. Thus, their primary purpose is to support and facilitate HiAP. Other approaches and tools, which are less specific to HiAP, can also be used to support it. These include inter-sectoral committees or teams, cross-cutting information, joint training, and integrated budgets [26,29,55]. These can help provide relevant evidence, facilitate collaborative working, and build a shared understanding of links to, and between, health and other sectors. HiAP uses many generic public health skills, such as the critical use of evidence and collaborating with stakeholders and communities. However, expertise in these technical skills is not sufficient [21,56]. A crucial part of the approach is engaging with policymakers and partners in other sectors and systems, and obtaining a better understanding of their specific jargon or language and constraints. 

## 4. Principles of a ‘Health in All Policies’ Approach

The principles and values that should guide HIA practice are well established, having been articulated in the Gothenburg consensus paper on HIA in 1999 [37] and updated by the International Association for Impact Assessment (IAIA), most recently in 2021 [47]. They reflect ethical principles that should inform wider decision making in public health [57] and are advocated in HIA guidance [38,47,58,59,60]. Based on these, we propose a set of principles that should underpin all health in all policies work, as shown in Box 1.

Box 1Principles underpinning Health in All Policies.
**Governance**—HiAP approach aims to foster accountability and shared social responsibility for health and well-being. It facilitates and promotes transparency about the health implications of policy decisions.**Comprehensive**—HiAP adopts a holistic approach to health. Rather than focusing on single health issues, it involves consideration of the range of health issues associated with each policy area or proposal.**Collaboration**—HiAP builds partnerships with colleagues in other sectors. It seeks to identify ‘win–wins’ that support the priorities of the policy area and also benefit health and health inequalities.**Equity**—HiAP considers not only overall health, but the distribution of health impacts across populations. It aims to reduce inequalities and prioritize the needs of populations with the poorest health.**Participation**—HiAP includes engagement with affected stakeholders and populations, and seeks to ensure that their views are taken into account in developing policy recommendations.**Evidence-based**—HiAP is based on the robust use of best available evidence, data and intelligence from different disciplines, to understand links between the policy area and health.**Sustainability**—HiAP considers impacts for both present and future generations. It seeks to balance environmental, social and economic impacts, and contribute to meeting the United Nations sustainable development goals.


## 5. Resources and Skills

Public health professionals working on HiAP require a broad range of skills. These include the skills to critically apply different kinds of evidence and data to appraise links between a policy area and health, and the skills to understand policy processes and opportunities [61]. Many of these are generic public health skills, such as the ability to apply a wide range of quantitative and qualitative data, health intelligence, and other evidence, to inform and influence decisions. They also need knowledge and understanding of specific processes and tools, such as HIA. An abundance of resources supports these technical aspects of HiAP. There is a plethora of toolkits, guidance, and resources for HIA [48,58,59,60,62,63], but, currently, there is less guidance available for HLAs. Training and resources are also available to support inter-sectoral collaboration, systems working, and to identify strategies to implement HiAP [52,64,65].

These technical skills may be considered the ‘science’ of HiAP, but public health professionals also need to gain tactical skills in the ‘art’ of HiAP [21]. The most important skill required is collaborative and partnership working. Public health professionals need to engage with policymakers in a productive, understanding and collaborative manner, in order to contribute constructively at different stages of policymaking. This may involve learning sector-specific terminology, producing joint resources, or holding joint events and capacity building training [65,66,67]. They need to develop an understanding of each other sectors’ perspectives, language and terminology, reference frameworks and decision-making processes, the constraints they may face in influencing health and inequalities, and the extent to which they can achieve this. This knowledge is often developed over time, through working in partnership [43], but better knowledge, understanding, and use of policy theories is also needed for HiAP, to realize its potential to achieve healthier public policies [30,68].

## 6. Mapping Implementation Activities for HiAP

Countries and states have taken different approaches to implementing HiAP. Finland introduced the concept during its 2006 Presidency of the EU. This built on decades of inter-sectoral work that focused on high-priority issues, and developed into an approach that integrates health into decision making across sectors. Mechanisms to support this include cross-sector committees, capacity building (both through formal training sessions and informally by ‘doing’ HiAP), and mandatory social and health impact assessments of proposed laws [14]. Other Scandinavian countries have also established high-level support for HiAP, with implementation at the local level by municipal staff [69]. The South Australian government introduced HiAP in 2007, with formal endorsement by the state cabinet. HiAP was closely linked to the South Australian strategic plan, with shared governance between the cabinet and government’s health department. A dedicated HiAP unit was established within the state government, to carry out health lens analyses of the prioritized policy areas [15]. California set up a HiAP task force in 2010, involving 22 state departments or agencies [70]. The task force has developed sectoral action plans, and has a dedicated team whose role includes supporting collaboration and embedding consideration of health and equity into the development and implementation of policies across agencies. Local government health departments across the USA have also adopted HiAP approaches [36]. In Wales, the well-being of the future generations (Wales) act 2015 [71] sets out a sustainable development-focused well-being agenda, which places integration, long-term thinking, prevention, collaboration, and involvement at the center of all decision making within public bodies in Wales. These ‘five ways of working’ intend to apply HiAP thinking (although implicitly and not by name), to maximize seven well-being goals, which include health, equity, economy, environment, and society, and provides a way to do so. Public service boards (PSBs) at a local level were established to collaborate and carry out local well-being assessments to identify local core needs and priorities, draft joint well-being plans to address these, oversee them, and share objectives and resources to achieve them. The approach is also supported by government long-term strategies, such as ‘Prosperity for All’, the recent ‘Programme for Government’ [72,73], and the Public Health (Wales) Act 2017 [74], which requires HIA to be statutory for public bodies in Wales in defined circumstances. The Wales Health Impact Assessment Support Unit (WHIASU), based in the national Public Health Institute for Wales, Public Health Wales, provides supportive resources, advice, and assistance for these HIAs across public health and other systems, for example, spatial planning or trade, in order to mobilize HIAP and foster cross-sector working [75].

There are many other examples of HiAP being used to inform individual decisions or policy areas. For example, in Scotland, the Scottish Health and Inequalities Impact Assessment Network (SHIIAN) has promoted and supported HIA for two decades, with minimal dedicated resources [76]. There are Scottish examples of HIA and other engagement with sectors such as spatial planning [77] and housing [78,79], and the planning (Scotland) act 2019 now requires HIA of planning proposals [80]. There is no formal requirement for HIA or other approaches to HiAP in other sectors, but there is a strong culture of inter-sectoral collaboration. Public health professionals in Scotland are increasingly engaging with policymakers at national and local levels, to address the determinants of health [81]. There are many other examples of countries and regions where HiAP approaches have been used, but fewer have an overall HiAP governance structure that uses a systematic approach to prioritize the policies that are most relevant to health [81,82]. A survey of 41 jurisdictions, including national, subnational and local governments, classified 13 in which HiAP practice was ‘established’, 10 as ‘progressing’, and 18 as ‘emerging’ [81].

## 7. Experiences of HiAP

Evaluations of the experiences of HiAP highlight several important pre-requisites and facilitators, as noted, for example, when considering the health impact of planning policies in urban cities [6]. Firstly, implementing HiAP across sectors in a systematic way requires political will, a long-term vision, and high-level commitment [14]. A review of the implementation of HiAP at local levels found that national leadership was ‘critical for successful and sustained HiAP’ [83]. Countries and states have established a clear mandate for HiAP, through cross-sectoral strategies, plans, or legislation [14,15,71], and through inter-sectoral structures that set priorities and maintain commitment and oversight [14,15]. The structures and mechanism used to do this may depend on local context, history, and culture. 

Secondly, HiAP needs to be resourced [83]. A key resource is public health professionals with dedicated time, capacity, and skills to understand and engage with other policy areas. However, teams to do so may be very small, have competing priorities, be funded in short-term political cycles, and dedicated units, such as those established in Wales [38] and South Australia [15,84], are rare. Where they do exist, there can be a tension between their role in capacity building, for example, to train the wider public health workforce in HIA, and reliance on such units to ‘do’ HiAP in practice. There is also a risk that resources that are not ‘ring fenced’ for such activity can be diverted to other, more immediate priorities and needs. Thirdly, HiAP also requires information resources, including data and evidence from a range of sources [14]. Finally, central to HiAP is collaborative working, which requires a high level of trust to be developed between public health and other partners [15]. Strong working relationships are important to help public health professionals develop their understanding of other policy areas, and also to help policymakers develop a shared vision and holistic understanding of how their work affects health [14,83,85]. This can support the implementation of recommended changes, and also influence policymakers’ future decisions and actions [44,45].

## 8. Challenges to HiAP Implementation

There are significant challenges to the implementation of HiAP, from both public health and other stakeholders [86,87].

A critical challenge is that HiAP is, by its nature, political, and may challenge some policy proposals. Although the focus is on identifying ‘win:wins’ and co-benefits [62,88], sometimes there is a conflict between health and other outcomes [58]. There may be a need to balance health gains against economic growth or other policy aims. HiAP may facilitate mature working relationships that enable trade-offs to be discussed and debated openly, but cannot completely avoid these conflicts. Where political priorities change, commitment to HiAP can be difficult to sustain, particularly where its focus on health equity challenges the prevailing ideologies [89,90].

HiAP has been criticized for promoting ‘health imperialism’, in seeking to prioritize health above other valid outcomes [30]. Public health stakeholders could be viewed as ‘interfering’ in other sectors. However, the counter-argument is that population health and well-being is a legitimate policy aim, which can enhance other outcomes, for example, a healthier and more productive workforce, so public health stakeholders should not apologize for promoting it. ‘Health imperialism’ should also be distinguished from ‘health sector imperialism’, which can narrowly focus on health care service delivery, and should not be used as a reason to dilute efforts to enhance population health and well-being [91]. Indeed, when viewed through the concept of the wider determinants of health, good health and well-being is indeed ‘everyone’s business’. Evaluations suggest that the fear of ‘health imperialism’ is overstated [30], and, instead, that health interests can be diluted by power imbalances, particularly if the focus on ‘win:win’ solutions inhibits more challenging discussions [71,92]. One response to the concerns about the perceived ‘health imperialism’ is to remove the word ‘health’ and instead consider the impacts of policies on ‘well-being’, which may be considered a concern not just of the health sector [93].

Another similar response is to integrate the consideration of health into other assessments, processes, or approaches [83,94]; for example, ex ante strategic environmental assessment under the ESPOO convention [95,96], or by taking an integrated approach to implementing the United Nations sustainable development goals (UN SDGs) [6]. However, these also bring challenges for HiAP. These may provide benefits by removing the element of ‘health imperialism’ and broadening the scope by holistically considering the impact on a wide range of sectors and goals, including health and well-being, whilst at the same time also engaging with key stakeholders, in a similar way to HiAP. However, these approaches do not have a primary focus on health impact and health equity. This means that the consideration of health could be diluted, or be subsumed by other issues. The potential for difficult negotiations also remains when trying to influence a policy that is likely to have adverse effects on health or well-being. Public health professionals need to work constructively and collaboratively (the whole point of HiAP) with other sectors, and avoid being overly critical, but also recognize that, at the same time, they may need to challenge policies that are likely to damage health [85,97]. There may be times when different stakeholders and partners cannot reach a common consensus, and public health practitioners need to explicitly oppose a policy proposal that is likely to cause health harm. This should be uncommon if partners are committed to working constructively together. 

It may be difficult to identify the policy areas and levels at which HiAP could achieve the greatest benefits, and to determine the most appropriate approach(es) required for each case. More comprehensive assessments that collate more evidence may reduce the uncertainty about the likely impacts, but smaller-scale inputs to decision making at the right time, for example, at the start of the policy making cycle, may be more influential. Only a few jurisdictions, such as South Australia, have a governance structure set up to prioritize policy areas for HiAP work, so examples of HiAP practice are often opportunistic. Whilst some evaluation has been carried out to date, in relation to the effectiveness of HiAP in influencing population health [84], more examples and evaluations of experiences of HiAP are needed to increase the knowledge and understanding of where HiAP is most valuable and what is required to support it in different circumstances. 

Demonstrating the impact of HiAP and component processes, such as HIA and HLA, is challenging. This reflects both a lack of monitoring of the health outcomes of policies once they have been implemented [98,99], and the difficulty of evaluating the impact of HiAP on policy decisions. Policy influence is not linear, but iterative and complex, and it may be difficult to disentangle the effect of an HIA or other public health input from other influences on a policy and the outcomes. Where public health professionals are involved from an early stage of policymaking, their impact on the final policy may be greater, but paradoxically less visible. The resultant changes in health, well-being, and equity may be difficult to track or attribute to any one policy in the long term [58,80]. This can make politicians and policymakers reluctant to support activity that may not reap rewards within a short political cycle or window, but over the long term—when they may no longer be in power. This can present difficulties in obtaining political buy-in and support and resources to increase capacity; therefore, public health needs to continue to increase HiAP awareness [54]. It also makes it hard to align policy cycles and co-ordinate ‘windows of opportunity’ to influence them. Even within the public health community, it can be difficult to prioritize work with a long-term focus on social determinants and future health inequalities, particularly when faced with immediate pressures, such as the COVID-19 pandemic or other health emergencies. Resources and capacity are finite, and public health organizations and institutions may need to cease some other work in order to support HiAP, which, again, is hard to untangle and can lead to difficult conversations about cost/benefits and the evaluation of its effectiveness. A lack of institutional resources can also have an impact on the provision of HIA or HLA training and capacity building [54], and may hamper data gathering, data sharing, or the monitoring of HiAP activity as work streams progress or policy changes lead to a different policy foci or emphasis [98,99,100].

Because of these issues, there could be a temptation for public health professionals to abandon the difficult task of influencing policy and instead be drawn into shared projects or interventions. These may be viewed as a safer and less challenging form of inter-sectoral work. They may generate more immediate, often high-profile actions, which are easier for politicians to support, but are much less likely to address the fundamental social determinants of poor health and health inequalities, such as poverty and racism [87,101].

## 9. Policy Support and Context for HiAP

Despite the challenges noted above, there are significant opportunities for the HiAP approach to contribute to national and international goals. The entry points to develop HiAP vary in different contexts, globally and nationally. The drivers in settings where HiAP has been introduced include an identified need to address the social determinants of health and inequalities in health, recognition of the need for public health to work with partners beyond the health sector, and commitment to ‘whole of government’ and ‘whole of society’ approaches [81,102].

Many governments now recognize the inter-connections between the aims of different policy areas, and explicitly prioritize the well-being of their citizens. The governments of Scotland, New Zealand, Iceland, Wales, and Finland are all members of the well-being economy governments partnership, adopting a ‘shared ambition of building well-being economies’ in which ‘policy is framed in terms of human and ecological well-being, not simply economic growth’ [103]. Globally, the United Nations sustainable development goals [104] highlight the need to consider multiple inter-connected goals in a holistic way, and there is a significant overlap between the SDGs and social determinants of health, and this framework has been suggested as an alternative path to HiAP or a complementary approach [105]. These provide useful goal-orientated frameworks to prioritize action, but there is still potential for conflict between different goals. For example, policies designed to increase employment and reduce poverty could adversely affect environmental goals. HiAP mechanisms, such as HIA, can make these potential conflicts explicit, and help identify ways to mitigate them and reduce inequalities. The routine use of HIA or HLA to scrutinize and review policy proposals can be a powerful way to deliver SDGs and achieve well-being economies in an integrated co-beneficial way [29,83,106]. Whilst the SDGs are time-driven, with the aim of implementation by 2030, HiAP is still evolving and timeless. HiAP and tools such as HIA can provide a holistic approach to policies beyond the timescale of the SDGs [46,107].

The COVID-19 pandemic also demonstrates the need for an integrated ‘whole of government’ and ‘whole of society’ approach [54]. The pandemic highlights the multiple ways in which other sectors affect health. For example, deforestation facilitates animal–human virus transmission [108,109], global transport networks contributed to the speed of virus transmission, overcrowded housing increases transmission within households, and precarious employment prevents people from self-isolating [110,111,112]. Conversely, policies that are intended to protect populations from transmission can impact the social determinants of health, such as the economy, transport, and education, with wider impacts on health [44,45,101,108]. The direct impacts of COVID-19 disease are also exacerbated by co-morbidities associated with pre-existing social determinants. Both direct and indirect impacts fall disproportionately on disadvantaged groups of people who already have poorer health, increasing health inequalities. This has led to the pandemic being described as a ‘syndemic’, which is a synergistic set of problems associated with a ‘perpetuating configuration of noxious social conditions’ that combine to damage health and increase health inequalities [113].

All these inter-related impacts highlight the explicit need for integrated policymaking across sectors. For example, the direct and indirect effects of climate change on health determinants are already apparent, and include extreme weather events, food insecurity, air pollution, increased vector-borne diseases, and population displacement [114]. Populations in countries with poor infrastructure and the greatest pre-existing health needs are the most at risk [13,115]. Mitigating and adapting to climate change will require wider changes in social determinants, such as transport and energy. These responses could bring both co-benefits and further risks to health, and could impact positively or negatively on health inequalities, depending on how they are formulated and implemented. An integrated approach is needed to balance these impacts and protect the people who are most at risk. In the United Kingdom (UK), the impacts of the pandemic and climate change are further exacerbated by ‘Brexit’ (the informal term for the UK withdrawal from the European Union). These all have significant implications for population health in their own right, but also act synergistically and cumulatively, creating a huge ‘triple challenge’ [116]. Therefore, it is now even more important for public health professionals and agencies, such as public health institutes, to mobilize and promote HiAP approaches as a platform to engage with a wide range of sectors and consider the population ramifications across society as a whole.

## 10. Conclusions—The Time Is Now

Whilst the concept of HiAP, and the use of HIA and related processes are not new, there is an urgent need to use these much more strategically and explicitly, both nationally and locally. The health and equity implications of policies outside of the health sector, such as economic development and planning, have long been recognized [1,7]. However, the COVID-19 pandemic and climate emergency further highlight the intertwined nature of impacts and policy responses. HiAP and the tools to apply it, provide a way to understand the breadth of the impacts that are primarily affected, and can deliver the SDGs and other related sustainability frameworks in an integrated way. The concept provides a vehicle through which to drive a sustainable, greener, more equitable and healthier recovery from the current public health emergency and any future events, such unexpected events at the international, national and sub-national level.

HiAP can be implemented successfully [49], with some countries using HIA or HLA systematically and effectively [39,41,43,45,84]. HiAP exemplifies the ‘art and science’ of public health by requiring both technical and tactical skills [21]. It requires public health professionals to invest the time to build partnerships and engage meaningfully with policies that affect the social determinants of health and health equity. A greater challenge is to gain, and sustain, the political commitment and momentum to support this approach systematically and implement policy changes [117]. Global commitment to the sustainable development goals and the example of well-being economy governments show that there is an appetite for more-integrated policymaking that centers around the well-being of people. HiAP gives us powerful mechanisms to achieve that aim, and they need to be mobilized now.

## Figures and Tables

**Figure 1 ijerph-18-09468-f001:**
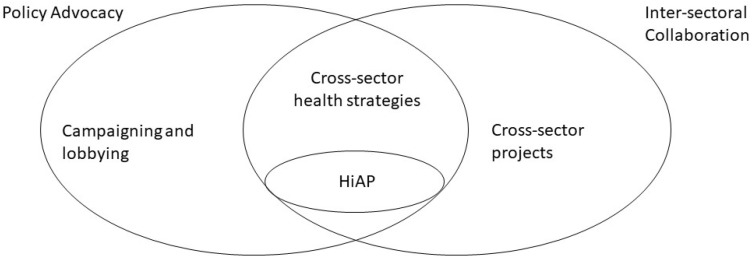
HiAP, policy advocacy and inter-sectoral collaboration (adapted from [35,36]).

**Table 1 ijerph-18-09468-t001:** HIA of unconventional oil and gas in Scotland [40].

Policy background and timing	In 2015, the Scottish Government agreed a moratorium on unconventional oil and gas (UOG) extraction in Scotland, pending a series of reviews to inform a decision about future policy.
HIA steps:	
1. Screening to determine whether to complete an HIA	Scottish Government requested an HIA to be carried out as part of the evidence to inform its policy.
2. Scoping the boundaries of the assessment—timeframes, resources, key stakeholders to engage with, evidence collection methods and key determinants and populations of focus	Scottish Government set initial terms of reference and timescale for the work, which was to address the following: Risks to health; Wider health implications; Potential mitigation of adverse impacts. Stakeholder workshops scoped relevant health issues to include in the review.
3. Appraisal of evidence, which is triangulated and analyzed	Evidence included the following: Workshops with community, industry and professional stakeholders to identify relevant impacts; Systematic review of published research on environmental hazards, pathways of exposure and association between hazards and health; Review of regulatory system and best practice.
4. Recommendations and reporting to inform decision makers	The HIA recommended a precautionary approach to UOG extraction in Scotland and made recommendations relating to the following: Future research; Community engagement; Use of HIA for UOG developments; Planning and regulatory systems; Monitoring and evaluation of UOG. A detailed HIA report, supplementary appendices and summary were provided to Scottish Government and published online.
5. Review and reflection including monitoring and evaluation of the process, impact/effectiveness and outcomes [39].	The HIA was subject to peer review before being finalized.The HIA was considered together with a wide range of other evidence. Both supporters and opponents of UOG quoted the HIA in their consultation responses. In 2019 the Scottish Government determined that UOG development should not be permitted in Scotland.

**Table 2 ijerph-18-09468-t002:** HLA of regional migrant settlement in South Australia [53].

Policy background and timing	The South Australian government had a target to increase inward migration in order to maintain population size.The HLA project aimed to develop understanding of links between settlement and health of migrants in order to develop policy responses.
HLA steps:	
Engagement with a wide range of key stakeholders, establishing relationships and connections to agree a policy focus between health and other sectors	The project team comprised staff from Department of Trade and Economic Development, Multicultural SA, and SA Health. They engaged with wider academic and other stakeholders to gather evidence and agree policy recommendations.
2. Evidence gathering to support and identify the impacts between health, well-being and the policy of study	Evidence included the following: Data on settlement patterns; Literature review on migrant settlement issues; Workshops with service providers; Focus groups with migrants and community These sources showed the interaction between social economic and health factors that affected health and other outcomes for migrants
3. Generating policy recommendations and a report	The project report made recommendations including the following: Accessible English classes for migrants; Training in use of interpreters; Funding for events to promote community inclusion. The team developed the migrant settlement well-being framework to inform data and information systems.
4. Navigating and steering the implementation of recommendations through decision-making processes in an effective way.	The recommendations were approved and adopted by the three government departments involved in the project team.
5. Evaluate the effectiveness of the process [49].	Evaluation of the HLA-involved group and individual interviews with key informants and a review of the project report.

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
