# Peer review of "‘Health in All Policies’—A Key Driver for Health and Well-Being in a Post-COVID-19 Pandemic World"

_ijerph, 2021, doi:10.3390/ijerph18189468_

Round 1

Reviewer 1 Report

I appreciate the opportunity to review this interesting piece. It does a good job explaining for a lay audience the concept of Health in All Policies (HiAP). I would not say that it is an original research piece, but more of a good explainer. I would be happy to assign a piece like this in the health policy course. The manuscript is well written and is apparently already formatted for publication, so I am unsure as to what changes can be made. The main part where I would appreciate revisions is in the descriptions of HiAP mechanisms (section 3). The authors present the HIA and HLA approaches and provide brief descriptions. They also mention that there are examples of these in practice, which are referenced in rapid form later in the manuscript (Section 6). It would be helpful in Section 3, however, to walk through two vignettes that demonstrate how each approach has been employed in practice. That would help the reader understand what each of the approaches looks like in practice and their similarities and differences.

Author Response

Reviewer 1

I appreciate the opportunity to review this interesting piece. It does a good job explaining for a lay audience the concept of Health in All Policies (HiAP). I would not say that it is an original research piece, but more of a good explainer. I would be happy to assign a piece like this in the health policy course. The manuscript is well written and is apparently already formatted for publication, so I am unsure as to what changes can be made.

Thank you for these comments.

The main part where I would appreciate revisions is in the descriptions of HiAP mechanisms (section 3). The authors present the HIA and HLA approaches and provide brief descriptions. They also mention that there are examples of these in practice, which are referenced in rapid form later in the manuscript (Section 6). It would be helpful in Section 3, however, to walk through two vignettes that demonstrate how each approach has been employed in practice. That would help the reader understand what each of the approaches looks like in practice and their similarities and differences.

Thank you for these suggestions. We have added two example case studies in section 3 and some further text to clarify the similarities and differences between HIA and HLA.

Reviewer 2 Report

This paper is not a "Review" in the scientific sense. It may be acceptable as a "Commentary" but does not present an original piece of research, i.e., lacks a clearly defined research question and/or testable research hypothesis and fails to report on any transparently described methodology. The authors are encouraged to review internationally recognized typologies for reviews and associated methods (e.g., as described in "A typology of reviews: an analysis of 14 review types and associated methodologies" https://doi.org/10.1111/j.1471-1842.2009.00848.x). Minimally, the manuscript should be extensively revised (1) using the IMRaD (Introduction, Methods, Results, and Discussion) format for scientific articles, (2) clearly identifying the knowledge gap justifying the need for a review, the present review type and its associated methodology, and (3) following an established reporting guideline for review articles, with the appropriate completed checklist (per the EQUATOR "Enhancing the QUAlity and Transparency Of health Research" standards, https://www.equator-network.org notably for systematic reviews). 

Author Response

Reviewer 2

This paper is not a "Review" in the scientific sense. It may be acceptable as a "Commentary" but does not present an original piece of research, i.e., lacks a clearly defined research question and/or testable research hypothesis and fails to report on any transparently described methodology. The authors are encouraged to review internationally recognized typologies for reviews and associated methods (e.g., as described in "A typology of reviews: an analysis of 14 review types and associated methodologies" https://doi.org/10.1111/j.1471-1842.2009.00848.x). Minimally, the manuscript should be extensively revised (1) using the IMRaD (Introduction, Methods, Results, and Discussion) format for scientific articles, (2) clearly identifying the knowledge gap justifying the need for a review, the present review type and its associated methodology, and (3) following an established reporting guideline for review articles, with the appropriate completed checklist (per the EQUATOR "Enhancing the QUAlity and Transparency Of health Research" standards, https://www.equator-network.org notably for systematic reviews). 

Thank you for your comments. The paper is not reporting a systematic review so we have not made these suggested changes.  

We clearly state in the manuscript and acknowledge that this is a narrative review and it follows the structure of similar previously published papers of this type in IJERPH for example, https://www.mdpi.com/1660-4601/18/12/6406/htm and https://www.mdpi.com/1660-4601/18/12/6330.

Considering the comments received from yourself and the academic editor, we would be willing for the paper to be published as either a narrative review, or a viewpoint.

Reviewer 3 Report

Review of “Health in All Policies’ – a key driver for health and well-being in a post COVID-19 pandemic world”

General:

The review paper provides a general and descriptive overview of HiAP and argues for its role in a COVID-19 context. However, the review doesn’t provide any convincing arguments for why HiAP approach and not the SDG approach should provide a key policy-making approach for governments in a post COVID-19 world, which is quite surprising considering the urgency to hinder climate change. The SDG approach is an integrated approach, which provides a lens for holistic policy-making. The paper could provide a structure of how HiAP could complement and be integrated in the SDG approach, since health, social and economic equity and environmental sustainability can be argued to have equally footing in a post COVID-19 world.

Secondly, I believe the key weakness of the review relates to a poor conceptualization of how HiAP should be integrated in policy-making in order to make effective policy changes. While the paper describes how HiAP is currently being integrated in national policy-making, it lacks a critical discussion of strength and weaknesses and policy outcomes of national HiAp approaches. It doesn’t really

Thirdly, the paper doesn’t clearly show what makes HiAP different from the HLA process. I can’t understand from the paper how HiAP is different from HLA in a way that has a relevance from a policy-making or health outcome perspective.

Specific comments:

Sometimes the text is very unspecific and vague, which makes it hard to distinguish  HiAP from its related concepts, for example: line 133-134.

You state that “…in a HiAP approach the starting point and focus is not a public health issue but a policy area or specific proposed policy [34]”. Yet further down, it seems that the focus of HiAP is public health…”it aims to develop a holistic understanding of how the policy area is likely to affect the range of health determinants in order to develop policy that will gain the best overall health outcomes ”

Even if it’s correct to say that the focus of HiAP is not on public health but to develop a policy area- what exactly is meant by this? And from such a perspective, then what differs HiAP from policy-making as-usual and the available agenda-setting tools and processes that are currently at the disposal for policy-makers?  

Section. 6: Experiences of HiAP

This section lacks a critical reflection of how HiAP is implemented, strengths and weakness of solutions and a discussion of what would consist of a best practice approach. Although, there is some mentioning of barriers and facilitators at the lines 321-333, this could be elaborated much more with a contextual understanding of what have triggered different developments.

Section of 7: Challenges to HiAP implementation

This section lacks a more detailed and structured discussion of what the challenges and conditions are for HiAP, e.g. monitoring and surveillance, ex-ante assessments of an integrated fashion, policy coordination, collaborative infrastructure, training, policy evaluations etc.

Section 8:

I would welcome more concrete suggestions for how HiAP could be integrated in Well-being Economy Governments Partnerships and in the SDG approach. What would concretely be added? In my understanding the two latter approaches already integrate public health and equity.

Author Response

Reviewer 3

The review paper provides a general and descriptive overview of HiAP and argues for its role in a COVID-19 context. However, the review doesn’t provide any convincing arguments for why HiAP approach and not the SDG approach should provide a key policy-making approach for governments in a post COVID-19 world, which is quite surprising considering the urgency to hinder climate change. The SDG approach is an integrated approach, which provides a lens for holistic policy-making. The paper could provide a structure of how HiAP could complement and be integrated in the SDG approach, since health, social and economic equity and environmental sustainability can be argued to have equally footing in a post COVID-19 world.

Thank you for your comments.

We believe the SDGs are a set of goals, rather than an approach. We have added some text to section 8 and 9 to clarify how HiAP can  help achieve the SDGs:

‘These provide useful frameworks to prioritise action, but there is still potential for conflict between different goals. For example, policies designed to increase employment and reduce poverty could adversely affect environmental goals. HiAP mechanisms like HIA, can make these potential conflicts explicit and help identify ways to mitigate them. Routine use of HIA or HLA to scrutinize and review policy proposals can be a powerful way to deliver SDGs and achieve Wellbeing Economies in an integrated way’

Secondly, I believe the key weakness of the review relates to a poor conceptualization of how HiAP should be integrated in policy-making in order to make effective policy changes. While the paper describes how HiAP is currently being integrated in national policy-making, it lacks a critical discussion of strength and weaknesses and policy outcomes of national HiAp approaches. It doesn’t really

A detailed comparison of different national approaches to HiAP is beyond the scope of this paper, which is trying to give an overview of what HiAP is. However we have added some further text to elaborate the section on pre-requisites for HiAP in section 6.  We have also added evidence comparing HIA and HLA approaches in section 3.

Thirdly, the paper doesn’t clearly show what makes HiAP different from the HLA process. I can’t understand from the paper how HiAP is different from HLA in a way that has a relevance from a policy-making or health outcome perspective.

HLA is one mechanism and practical tool, but not the only one, that can be used in a HiAP approach to policy making. The case studies in section 3 also elaborate on this further.

Specific comments:

Sometimes the text is very unspecific and vague, which makes it hard to distinguish  HiAP from its related concepts, for example: line 133-134. You state that “…in a HiAP approach the starting point and focus is not a public health issue but a policy area or specific proposed policy [34]”. Yet further down, it seems that the focus of HiAP is public health…”it aims to develop a holistic understanding of how the policy area is likely to affect the range of health determinants in order to develop policy that will gain the best overall health outcomes ”

Even if it’s correct to say that the focus of HiAP is not on public health but to develop a policy area- what exactly is meant by this? And from such a perspective, then what differs HiAP from policy-making as-usual and the available agenda-setting tools and processes that are currently at the disposal for policy-makers?  

We have amended the text to clarify the point we were trying to make. It now says ‘Unlike the approaches discussed above, in a HiAP approach the starting point and focus is not a single public health issue but a policy area or specific proposed policy [34]. For example a traditional public health approach may start with a problem like physical inactivity and seek to work with a range of partners, such as transport policymakers, whose policies might influence physical activity levels. On the other hand, a HiAP approach starts with a policy area like transport policy. It then aims to develop a holistic understanding of how the policy area may affect not only physical activity but a range of health determinants, in order to develop policy that will gain the best overall health and equity outcomes. This is a crucial difference. It means that HiAP work requires a more detailed understanding of the constraints and opportunities of the relevant policy area. Strong working relationships between public health and colleagues in the other policy ar-ea are important to facilitate this. It also requires specific mechanisms or tools to identify the range of potential links with health.’

Section. 6: Experiences of HiAP

This section lacks a critical reflection of how HiAP is implemented, strengths and weakness of solutions and a discussion of what would consist of a best practice approach. Although, there is some mentioning of barriers and facilitators at the lines 321-333, this could be elaborated much more with a contextual understanding of what have triggered different developments.

A detailed comparison of different national approaches to HiAP would usefully form another more technical paper, whereas the current paper aims to give an overview of what HiAP is and overall principles and experiences of practitioners. We want to set the scene and a provide a future outlook and perspective for the implementation of HIAP through the use of tools such as HIA. However, we have added some further text to elaborate the section on pre-requisites for HiAP in section 6.  We have also added evidence comparing HIA and HLA approaches in section 3.

Section of 7: Challenges to HiAP implementation

This section lacks a more detailed and structured discussion of what the challenges and conditions are for HiAP, e.g. monitoring and surveillance, ex-ante assessments of an integrated fashion, policy coordination, collaborative infrastructure, training, policy evaluations etc.

We believe the conditions for HiAP are noted in the pre-requisites section in section 6, and the need for training and other resources are addressed in section 5. We have also proposed the Principles in Section 4 that underpin HiAP and are elaborated further throughout the text. However, we have have also expanded the text in section 7 (now 8) and briefly highlight the issues.

Section 8:

I would welcome more concrete suggestions for how HiAP could be integrated in Well-being Economy Governments Partnerships and in the SDG approach. What would concretely be added? In my understanding the two latter approaches already integrate public health and equity.

We have added some text to clarify this further: ‘These provide useful frameworks to prioritise action, but there is still potential for conflict between different goals. For example, policies designed to increase employment and reduce poverty could adversely affect environmental goals. HiAP mechanisms like HIA, can make these potential conflicts explicit and help identify ways to mitigate them. Routine use of HIA or HLA to scrutinize and review policy proposals can be a powerful way to deliver SDGs and achieve Wellbeing Economies in an integrated way’.

Round 2

Reviewer 2 Report

I do not believe this article meets the standards of a scientific research paper, with a transparently described methodology. It is not suggested that a systematic review protocol (which may include narrative synthesis among its analytical techniques) needs to have been followed, but that the review type and its associated methodology (including the search strategy and search results) be clearly identified. Much of the text reads as a commentary rather than a research paper. As also mentioned by one of the other reviewers, “I would not say that it is an original research piece, but more of a good explainer.” The authors are entitled to their opinion that methodology and unbiased reporting of how they chose their sources to review matter not.

Author Response

Please find below the authors response to Reviewer 2: 

Thank you for your comments. We have published the manuscript as a Viewpoint (commentary) paper as was suggested in your comments. 

We strongly believe that transparent methods and processes are super important alongside robust and traceable evidence. However, the purpose of the paper was not intended to be a systematic review as we have stated. We feel that a systematic review would have been inappropriate for our purposes as we did not set out to answer a single research question but to give a broad overview of HiAP, including the mechanisms, experience, principles to underpin it and its relevance in a post pandemic context.